# Designing a computer-assisted diagnosis system for cardiomegaly detection and radiology report generation

**Tianhao Zhu[1,2], Kexin Xu[1,¤a], Wonchan Son[1,¤b], Kristofer Linton-Reid[2], Marc Boubnovski-Martell[2], Matt Grech-Sollars[2,3,4], Antoine D. Lain[1]\*, Joram M. Posma[1]\***

**1** Section of Bioinformatics, Department of Metabolism, Digestion and Reproduction, Imperial College London, London, United Kingdom, **2** Department of Surgery and Cancer, Imperial College London, London, United Kingdom, **3** Department of Computer Science, University College London, London, United Kingdom, **4** Lysholm Department of Neuroradiology, National Hospital for Neurology and Neurosurgery, University College London Hospitals NHS Foundation Trust, London, United Kingdom

¤ a Current address: MRC WIMM Centre of Computational Biology, Radcliffe Department of Medicine, University of Oxford, Oxford, United Kingdom
¤ b Current address: MEDICAL IP, Seoul, South Korea
* a.lain@imperial.ac.uk (ADL); jmp111@ic.ac.uk (JMP)

**Funding:** This research was funded by Health Data Research (HDR) UK and the Medical

## Abstract

Chest X-ray (CXR) is a diagnostic tool for cardiothoracic assessment. They make up 50% of all diagnostic imaging tests. With hundreds of images examined every day, radiologists can suffer from fatigue. This fatigue may reduce diagnostic accuracy and slow down report generation. We describe a prototype computer-assisted diagnosis (CAD) pipeline employing computer vision (CV) and Natural Language Processing (NLP). It was trained and evaluated on the publicly available MIMIC-CXR dataset. We perform image quality assessment, view labelling, and segmentation-based cardiomegaly severity classification. We use the output of the severity classification for large language model-based report generation. Four board-certified radiologists assessed the output accuracy of our CAD pipeline. Across the dataset composed of 377,100 CXR images and 227,827 free-text radiology reports, our system identified 0.18% of cases with mixed-sex mentions, 0.02% of poor quality images ($F1 = 0.81$), and 0.28% of wrongly labelled views (accuracy 99.4%). We assigned views for 4.18% of images which have unlabelled views. Our binary cardiomegaly classification model has 95.2% accuracy. The inter-radiologist agreement on evaluating the generated report's semantics and correctness for radiologist-MIMIC is 0.62 (strict agreement) and 0.85 (relaxed agreement) similar to the radiologist-CAD agreement of 0.55 (strict) and 0.93 (relaxed). Our work found and corrected several incorrect or missing metadata annotations for the MIMIC-CXR dataset. The performance of our CAD system suggests performance on par with human radiologists. Future improvements revolve around improved text generation and the development of CV tools for other diseases.

Research Council (MRC) via an UKRI
Rutherford Fund Fellowship to J.M.P.
(MR/S004033/1). ADL and JMP are additionally
supported by the Horizon Europe project
CoDiet. The CoDiet project is funded by the
European Union under Horizon Europe grant
number 101084642. CoDiet research activities
taking place at Imperial College London is
supported by UK Research and Innovation
(UKRI) under the UK government's Horizon
Europe funding guarantee (grant number
101084642). JMP is supported by the MRC
project GI-tools (MR/V012452/1). The funders
had no role in the design of the study; in the
collection, analyses, or interpretation of data; in
the writing of the manuscript, or in the decision
to publish the results.

**Competing interests:** The authors have
declared that no competing interests exist.

## Author summary

Chest X-rays (CXRs) are a key tool for diagnosing heart and lung conditions. Radiologists review hundreds of images a day, which can result in fatigue and reduced accuracy. In this study, we developed an AI-powered system to automate parts of the CXR analysis process. Using a large, public dataset (MIMIC-CXR) we created models that assess image quality, label the orientation (view) of the image, and classify the severity of an enlarged heart (cardiomegaly) as use-case. Our system also generates radiology reports by using a large language model (LLM) to generate human-like text based on the cardiomegaly severity. When tested, the system correctly labelled CXR views with 99.4% and identified cardiomegaly with 95.2% accuracy. Four experienced radiologists reviewed the LLM-generated reports and found they were nearly as accurate as reports from human radiologists. Our system also identified and corrected errors in the labels of the original dataset. Our system has not been deployed in clinic, as further evaluation and improvements are needed, however it demonstrated the potential of AI to aid radiologists and reduce their workload, act as second opinion, and speed up report generation.

## Introduction

For over a century, chest X-rays (CXR) have been the primary diagnostic tool in thoracic imaging due to their fast acquisition, low cost, and versatility [1,2]. Thoracic anatomical structures, such as the lungs and heart, are visualised and distinguished on a 2D image through radiodensity variations. This enables the detection of common cardiothoracic conditions such as pneumonia, pneumothorax and tuberculosis from a CXR image [3].

However, CXR has some limitations compared with computed tomography (CT) or magnetic resonance imaging (MRI). CXR reduces the 3D pleural space to a singular plane, leading to structural overlaps between anatomical features, with over 40% of the lung parenchyma obscured by the ribs and mediastinum [4]. Depending on the type and site of a lesion, the visible differences between the lesion and surrounding structures may be subtle or even indistinguishable [5,6].

Apart from technical limitations, human factors also complicate CXR interpretation. CXR's short acquisition time has led to widespread use as a primary diagnostic test. This resulted in the average US radiologist workload increasing by 30% to 70% in three decades, contributing to significant radiologist burnouts [2,7–9]. Combined with radiologist shortages, this causes a struggle to meet the demand for timely reports [10]. Radiologists are affected by cognitive biases, distractions, and fatigue, which negatively impact their diagnostic acumen [11,12]. Despite technical advancements and error-mitigating strategies over the past 75 years, radiology interpretive error rates have remained at 3% to 4% [11–13].

To address this, recent exploratory efforts aim to integrate artificial intelligence (AI) technology into CXR interpretation [5]. The underlying rationale is that 'AI readers', unlike humans, are not affected by fatigue and distractions. AI can avoid certain cognitive biases due to its training methods, leading to higher accuracy and consistency [5]. Furthermore, AI could reduce the workload of radiologists, thereby decreasing backlogs and improving patient outcomes [14,15].

While most AI approaches to CXR interpretation use deep learning methods, such as convolutional neural networks (CNNs) [5,16,17], they differ in their application area. The computer-assisted diagnosis (CAD) system CheXGAT [18] is an AI-augmented diagnostic

tool to help radiologists detect and discern between fourteen chest conditions. Whereas AI-CenterNet CXR [19] was developed as an automated system for detecting and localising eight thoracic conditions without human involvement. Other work focuses on developing AI-tools for detecting specific diseases [5,20], such as CNN-based models COVID-Net CXR-2 [14] and Covid-MANet [15] developed for detecting COVID-19 pneumonia.

Our study focuses on developing a proof-of-concept CAD system that integrates classification with report generation. We focus on the narrow task of cardiomegaly (abnormal or pathological enlargement of the heart) detection. Experienced radiologists can quickly identify severe cardiomegaly from a Posterior-Anterior (PA) CXR, while more ambiguous cases may require calculating the cardiothoracic ratio (CTR) using the same PA CXR (Fig 1). Any value above a threshold (often set between 0.45~0.55) possibly indicates cardiomegaly [21–23]. The process can be time-consuming and contain a degree of subjectiveness due to variations in thresholds used and measurements made, thus could be improved by using CAD systems.

Several CAD systems have been developed for cardiomegaly detection and can be categorised into classification-based and segmentation-based approaches [24]. A recent transfer learning model [24] achieved an Area-Under-the-ROC Curve (AUC) of 0.87 for classifying normal and cardiomegaly CXR images, in line with similar approaches [25–27]. Segmentation models based on the U-Net architecture directly calculate the CTR to derive binary decisions to classify cardiomegaly [24,28]. While these models claim higher explainability [29,30],

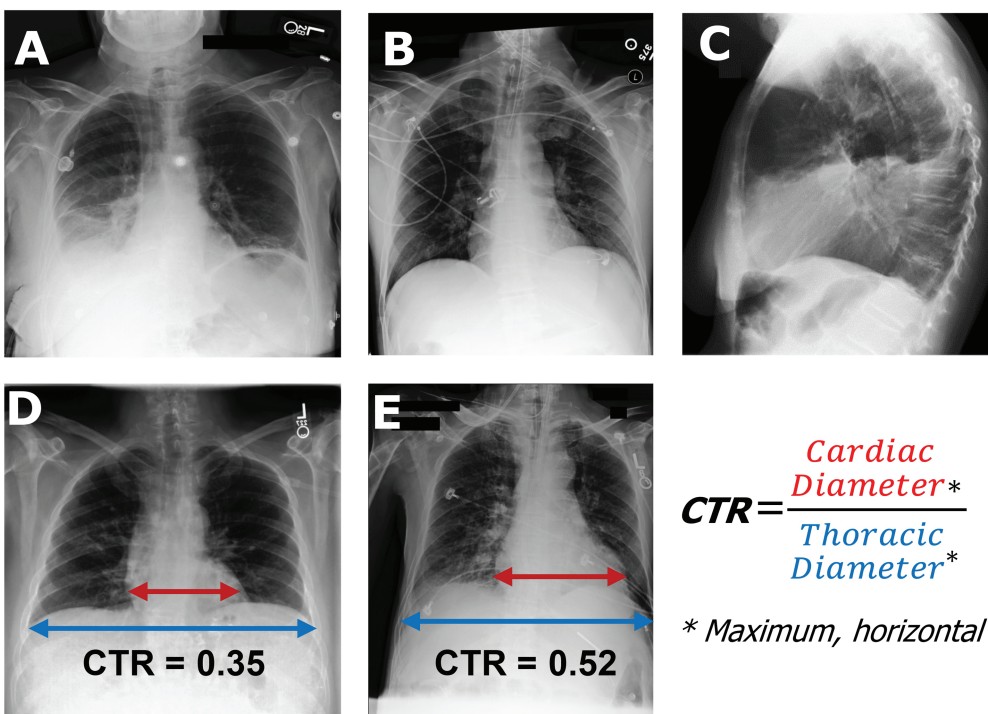

**Fig 1. Chest X-ray views and cardiothoracic ratio calculation from frontal images. (A)** Frontal (Posterior-Anterior, PA) chest X-ray image. (**B**) Frontal (anterior–posterior, AP) chest X-ray image. (**C**) Lateral chest X-ray image. (**D**) Frontal chest X-ray image with normal heart size with the two components of the cardiothoracic ratio (CTR) indicated. (**E**) Frontal chest X-ray image with enlarged heart size (CTR ≥ 0.5). (A, D, E) images from [31]. (B, C) images from [32].

they may not be as predictive due to the limitation of lack of pixel-level labelled CXR data, restricting model generalisability [24,30]. Recent advancements in natural language processing (NLP) and large language models (LLMs) can be leveraged to generate realistic report sentences that are styled as those written by human radiologists. Currently, without the ability to determine the severity of heart enlargement from machine predictions and without evaluating the quality of computer-generated reports by human radiologists, we cannot fully assess how effective these CAD systems are at automating cardiomegaly detection and reporting.

We hypothesise that coupling the output of computer vision (CV) models with LLMs can generate plausible expert-level radiology report texts. Specifically, our objectives in this study are to 1) assess the quality of the MIMIC-CXR dataset, including both images and reports, to prepare it for developing CV models, 2) utilise CXR segmentation algorithms to derive features relevant to severity classification, 3) combine the output of the CV classification model with large language models to generate plausible medical text, and finally 4) use expert evaluation of the generated reports to assess the quality of the generated text.

## Materials and methods

### MIMIC-CXR dataset

The MIMIC-CXR database [33] was selected as the primary source of CXR data in this study. It is publicly available with credentialed access via PhysioNet [34] and includes 377,100 CXR images in both JPG and DICOM format (https://physionet.org/content/mimic-cxr-jpg/2.0.0/) and 227,827 corresponding free-text radiology reports (https://physionet.org/content/mimic-cxr/2.0.0/), all anonymised. The images and corresponding text reports are stored across 10 folders (p10-p19). Additionally, the database contains CSV files with image metadata and feature annotations acquired with CheXpert [35] and NegBio [36]. In each feature column, 1 represents a positive mention of the feature, 0 represents negative mentions, and -1 represents ambiguous mentions. A blank value indicates the absence of any mention of the feature.

All CXR images in JPG format and their corresponding free text reports, alongside metadata and annotation CSV files, were directly downloaded from PhysioNet using command prompts.

The data user agreement of MIMIC-CXR prohibits sharing images from the dataset. Images from MIMIC-CXR that are already in the public domain also fall under this agreement. The MIMIC-CXR team have confirmed that this policy applies to all data therefore no disclosure is permitted (private communication 30 August 2024). We therefore have replaced the images with similar images from the NIH Chest X-ray dataset [31] (for frontal PA images) and from [32] (for AP and lateral images, under a CC BY-NC 3.0 license). In each of the figures (where relevant) we have indicated the original MIMIC ID that the inference is made on so that those with credentialised access to MIMIC-CXR can inspect the original data. The report snippets shown here are all synthetically generated, however their context mimics the original MIMIC-CXR report text. MIMIC-CXR report IDs are disclosed in the relevant captions to allow readers with MIMIC-CXR data access to inspect the original images and reports.

**Generation of poor quality images.** All 75,216 images from p10 and p11 folders were scrutinised manually to identify 'poor quality' images which are those that cannot be properly interpreted by radiologists and would warrant retaking the image. The 22 identified 'poor quality' images contain severe cropping, distortion, and/or absence of thoracic structures and were used to generate 500 additional 'artificial' poor quality images by applying synthetic and augmentation techniques to imitate common types of poor quality images from normal

images (see examples in Results). The 'genuine' poor quality images were set aside for testing (see below).

**Ethics statement.** As described in the original dataset description [33] this project was approved by the Beth Israel Deaconess Medical Center's Institutional Review Board. All data was anonymised and the data did not impact clinical care, therefore the requirement for individual patient consent was waived by the board.

## Computer vision models

Two classification models were trained on the MIMIC-CXR dataset, one to classify normal from poor quality images, and a subsequent model to classify frontal from lateral images as shown in Fig 2A to perform an image quality control.

**Classification.** For the normal versus poor quality image classification model, 500 frontal including both PA and AP views as well as 500 lateral images were selected at random for the normal group from the p10 and p11 folders to form the training set, where each image was manually checked.

For the frontal versus lateral classification model, 1,000 images from each group were selected from the entire MIMIC-CXR dataset (p10-p19) through cross-referencing with view labels reported in the metadata CSV file. Similarly, each image was manually checked for labels being correctly assigned. The test set consists of another 2,000 images (equally split) selected at random, and manually checked, without overlapping patients with the training set.

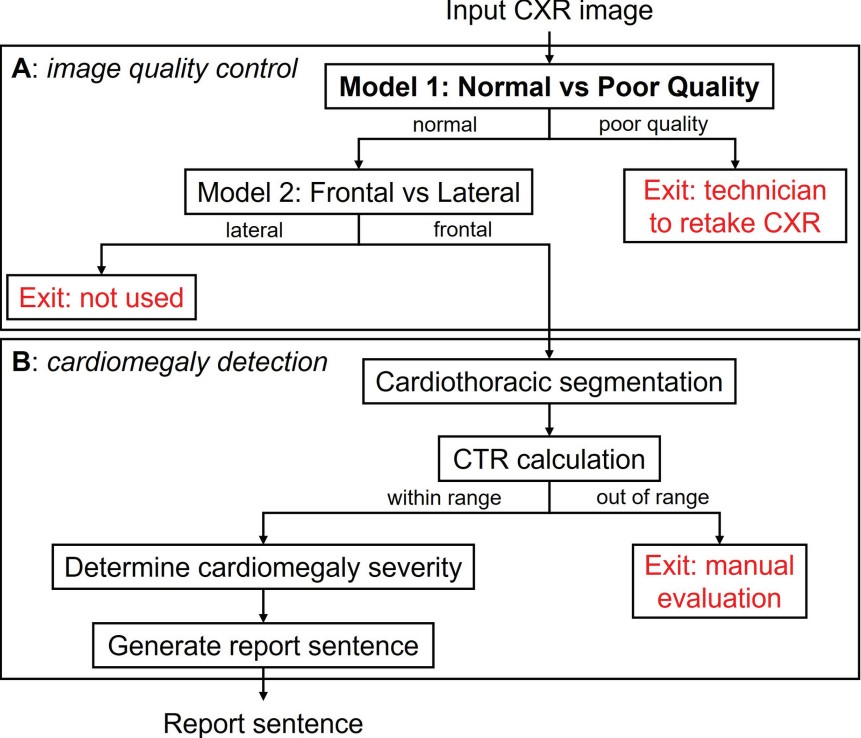

**Fig 2. Flow chart of the computer-assisted diagnosis pipeline.** (**A**) Image classification models. (**B**) Image segmentation and sentence generation models. Indicated in red are 'exit' steps where the image or result cannot be used by the pipeline.

For both models, the training image datasets were randomly split into training and validation sets at a ratio of 80:20 for model training performance monitoring. The ResNet50 model [37] was loaded without pre-trained weights, and all images were resized to $256 \times 256$ pixels. Data augmentation was incorporated into training data loading which randomly introduced contrast jitter and rotations to all images to increase data variability, model versatility and robustness. Multiple combinations of training parameters and hyper-parameters were experimented with for performance optimisation in a grid search-like process with manual tuning. For both training and validation sets, the batch size was set to 64, with cross-entropy loss and Adam optimiser (learning rate of 0.001). 20 epochs were scheduled, with an early stopping condition if both training and validation accuracy surpassed 99% at the end of an epoch. The trained model states were saved as .pth (PyTorch-trained model) files. Each model was initialised 10 times with different random states and results are represented as median with interquartile range.

The trained models were subsequently evaluated on the independent test sets. Performance was gauged by overall accuracy, as well as precision and recall, with confusion matrices constructed. To understand the rationale of the model in classifying images, Grad-CAM [38] was used to create attention heatmaps to visualise the focus areas of each model, providing insight into what features it considers distinctive to each class, as well as catching instances where the model may predict based on non-relevant features.

**Segmentation and cardiothoracic ratio calculation.** Good quality frontal CXR images with cardiomegaly mentions in their associated reports were then grouped based on the severity descriptors extracted from text reports. For each frontal image, segmentation of the lungs and heart was carried out using the inbuilt default U-Net-based segmentation model in the `Chest X-ray Anatomy Segmentation` (CXAS) package [39] (v0.0.9), generating lungs and heart masks in the PNG format. The segmentation model was trained via CT-CXR projection, using 10,021 thoracic CT with 157 structural labels projected onto a two-dimensional plane. Authors of the model claimed a high model-human annotator agreement, with mIoU scores of 0.93 and 0.85 for frontal and lateral anatomies respectively. Moreover, the authors emphasized the suitability of the model for use with extracting explainable medical features, for instance CTR.

The Maximal Horizontal Diameter (MHD) of the lungs and heart masks were used to calculate the computer vision-based CTR (CV-CTR). The MHD was calculated by counting the number of mask pixels in each row of a given mask image using `opencv-python` (v4.9.0.80) and `numpy` (v1.26.2) packages, and subsequently taking the 95[th] percentile value to eliminate potential outliers caused by sub-optimal segmentation. If either the lung or heart MHD of an image was 0, it was excluded from further analysis as segmentation had likely failed for that image. The CV-CTR for each image was then calculated by dividing the heart MHD by the lung MHD (Fig 1). The CV-CTRs were subsequently grouped by severity descriptors from the reports. Additionally, another set of ratios was calculated using the total areas, instead of HMD, of heart and lung masks. Areas were measured by counting the total number of mask pixels within lung and heart masks, and the Cardiothoracic Area Ratio (CAR) was calculated by dividing heart by lung area.

The code for classification models can be found at https://github.com/omicsNLP/CXR-Based-Cardiomegaly-CAD.

## Natural language processing tools

**Extracting data from reports.** CXR reports containing either positive or negative mentions of cardiomegaly and its synonyms were filtered by selecting only reports labelled 1 or

0 in the 'cardiomegaly' column in the CheXpert CSV, thereby rejecting any reports with ambiguous mentions or no mentions concerning cardiac size. For each selected report, NLP methods, including text tokenisation and keyword searching, were applied using the NLTK (v3.8.1) package to extract sentence(s) describing cardiomegaly or cardiac size. Cardiomegaly mentions were extracted from a total of 137,424 reports through this process, roughly 60% of all MIMIC-CXR dataset reports.

All adjectives and adverbs from extracted cardiomegaly mentions were identified and ranked based on frequency. From the top 50 most frequent adjectives and adverbs respectively, severity descriptors (terms that describe the severity of a condition) that appeared in at least 100 reports were selected for further analysis. Adjectives and adverbs sharing the same stem (such as mild and mildly) were merged. Cardiomegaly mentions files containing these severity descriptors were grouped by keyword matching, with any files containing two or more different severity descriptors being discarded.

The CV-CTR values within each group of adjectives were tested for normality using the Shapiro-Wilk test. Due to deviations from normality, Mann-Whitney U-tests (Wilcoxon rank sum test) were used to compare groups. The Šidák correction was used to adjust for multiple testing. Groups were combined when they are not significantly different. Threshold values separating adjacent groups were calculated separately for both CV-CTR and CAR. One-sample Wilcoxon signed rank tests were performed for every value between 0 and 1 at intervals of 0.01 with CV-CTR data (0 to 0.5 for CAR) of each severity group, and each value was assigned to the severity group with the least significant p-value, forming ratio intervals.

**Comorbidity analysis.** Comorbidity analysis was performed for each report using NLP annotations of 14 conditions present in the CheXpert CSV. The conditions were 'Atelectasis', 'Cardiomegaly', 'Consolidation', 'Edema', 'Enlarged Cardiomediastinum', 'Fracture', 'Lung Lesion', 'Lung Opacity', 'No Finding', 'Pleural Effusion', 'Pleural Other', 'Pneumonia', 'Pneumothorax', and 'Support Devices'. Any positive mention of the condition in the report (labelled '1' in the CSV) was considered an occurrence. Ochiai similarity (Eq 1) was calculated between conditions to analyse their degree of association (where $a$ is the number of co-occurrences of conditions A and B, $b$ the number of single occurrence of condition A, and $c$ the number of single occurrence of condition B). We used Ochiai similarity instead of co-occurrence frequency as different conditions had varying occurrence frequencies. The Ochiai similarities among the 14 conditions were used to construct a graph of co-occurrences using the NetworkX package (v3.1).

$$\text{Ochiai similarity} = \frac{a}{\sqrt{(a+b)(a+c)}} \qquad (1)$$

**Report text generation.** Previously extracted cardiomegaly mentions were regrouped with severity categories. The mentions were pre-processed to strip periods and all letters were converted to lowercase to account for minor variations in expressions. Within each severity category, the most frequently used sentences for cardiomegaly reporting were counted and ranked. Expressions that mention other conditions irrelevant to cardiomegaly and/or patient history were filtered out. These sentences were subtly altered by changing some words with synonyms (e.g. 'borderline' with 'top-normal', 'cardiac silhouette' with 'heart', etc.), ensuring no sentence could be traced to any individual report. The ten most frequent filtered expressions from each category were passed to MedQBot on Poe (https://poe.com/medqbot), an LLM originally intended for medical exam preparation, with a prompt to generate ten similarly concise, report-styled, cardiomegaly expressions. The prompt used is listed below, where [severity group] was replaced by each of the adjectives or adverbs identified above:

*Here are some CXR report sentences describing* [severity group] *cardiomegaly. Based on this set of sentences, generate similarly concise and report-styled sentences describing* [severity group] *sentences.*

The generated sentences were acquired on 13 March 2024. All sentences were manually checked for accuracy, style and relevance, with unsuitable sentences (e.g. too verbose, unrelated, inaccurate, temporal) being discarded.

## Human evaluation

To evaluate our cardiomegaly CAD pipeline performance, four radiologists (three with 1 to 5 years of experience, and one with 10+ years of experience) were invited to participate in a small-scale assessment using an online Qualtrics questionnaire. A total of 40 items were prepared, each consisting of a frontal CXR image, a report snippet of the findings section, and a side-by-side question box where the radiologists indicate to what degree they agree with each sentence from the snippet based on the image provided. The radiologists were able to provide reasoning for any disagreement in a free-text box. Each case was reviewed by two radiologists, who were blinded to each other's responses and unaware of whether the sentences were machine-generated or taken from the original MIMIC report. Each radiologist evaluated a similar number of sentences, with the same ratio of changed and unchanged sentences.

The questions were divided into four categories (Table 1) to ensure an even mix of original and machine-generated sentences, either related or unrelated to cardiomegaly. The machine-generated cardiomegaly sentences were randomly selected from the pre-curated list as described before based on the predicted severity group by the CV model. For question categories 2 and 4, non-cardiomegaly sentences from the 'findings' section underwent the same process as described for cardiomegaly above by word swapping with synonyms. The individual sentences were then passed in random order, breaking any relationships between augmented sentences from the same findings session, to MedQBot to be paraphrased using the following prompt on 19 March 2024:

*For the following paragraph taken from a CXR report, generate a new paragraph by paraphrasing each sentence from the given paragraph, while maintaining concision and standard CXR reporting style.*

Each radiologist was assigned 20 items (each containing multiple statements to assess), with each radiologist receiving roughly the same number of items in each listed category. The assignment process also ensured that each question was answered by exactly two radiologists. The radiologists were not informed of the design or pipeline of this study nor the composition of the question set or that of their assigned questions. Their responses were individually downloaded as CSV from Qualtrics and pre-processed to integrate into a single data

**Table 1. Question composition of radiologist questionnaire.** Where 'unchanged' indicates the sentences are those from the original MIMIC-CXR reports, 'CAD system' indicates cardiomegaly sentences are generated with the pipeline described above, 'MedQBot' indicates sentences generated by LLM-paraphrasing.

| Category | n | CM sentences | Other sentences |
|---|---|---|---|
| 1 | 8 | Unchanged | Unchanged |
| 2 | 8 | Unchanged | MedQBot |
| 3 | 12 | CAD system | Unchanged |
| 4 | 12 | CAD system | MedQBot |

frame using pandas (v2.1.4) with paired responses for each question, complete with question category and cardiomegaly relevance.

To assess inter-rater agreement we computed the total agreement by comparing their results for each sentence. We used two agreement metrics: strict agreement which consisted of four distinct categories ('Agree', 'Partially Agree', 'Disagree', and 'Unsure') and relaxed agreement which merged 'Agree' and 'Partially Agree' into a single category.

## Results

### Data quality assessment

We first reviewed the data quality for several aspects relevant to this study. The MIMIC-CXR dataset is one of the largest publicly available chest X-ray radiology datasets and includes considerable data variability. Consequently, only a subset of the MIMIC-CXR dataset is suitable for developing cardiomegaly detection CAD systems.

Selecting appropriate subsets of a dataset for downstream applications is complicated by many factors. As mentioned, frontal CXR images are best suited for cardiomegaly detection, while lateral images are rarely used and therefore were excluded. Furthermore, the MIMIC-CXR dataset is not error-proof. We found that 421 patient reports contain mismatched sex mentions (Table 2). View labels of images were occasionally missing or incorrect, such as mislabelling a lateral image as frontal. In addition, a smaller proportion of images have quality issues that completely prevent analysis. To address this issue, the CAD system must first identify and flag unsuitable images to avoid further processing and analysis. This is accomplished by leveraging two ResNet50 classification models as detailed below.

As mentioned, a two-step hierarchical model was developed to classify CXR images by quality and view for dual purposes. In direct relation to the CAD system itself, the model provides an evaluation step to only allow suitable images to be assessed by downstream cardiomegaly detection algorithms. Additionally, the model can also help correct mislabelling and identify flawed images in the MIMIC-CXR dataset (see Fig 3A for synthetic training images).

This model demonstrated an overall accuracy of 99.2% (AUC = 0.989, precision = 0.996, recall = 0.995) on a test set with 20 real poor quality images from the MIMIC-CXR and 1,000 normal images with 85% recall for poor quality class (Fig 3B). The second model, which differentiates between frontal and lateral CXR, exhibited better performance, with 99.4% accuracy (AUC = 0.996, precision = 0.993, recall = 0.996) on the test set with 1,000 frontal and lateral images each (Fig 3D).

**Table 2. Data quality assessment of the MIMIC-CXR dataset (65,379 patients, 227,827 individual reports, 377,100 images). Indication of mismatched sex mentions in reports attributed to the same individual, number (%) of poor quality images indicated by our poor quality image classification model, and number (%) of wrongly labelled views (in the metadata) indicated by our view classification model. All reports and images indicated above were manually checked, and we provide a spreadsheet in S1 Data with the corrected view labels and reports likely from different individuals due to sex differences with other reports attributed to the same person identifier.**

| Data quality assessment category | n (Total) | % |
|---|---|---|
| Patient reports with mixed sex mentions | 421 (227,827) | 0.18 |
| Poor quality images | 83 (377,100) | 0.02 |
| Wrongly labelled views | 1,054 (377,100) | 0.28 |
| Unlabelled views | 15,769 (377,100) | 4.18 |

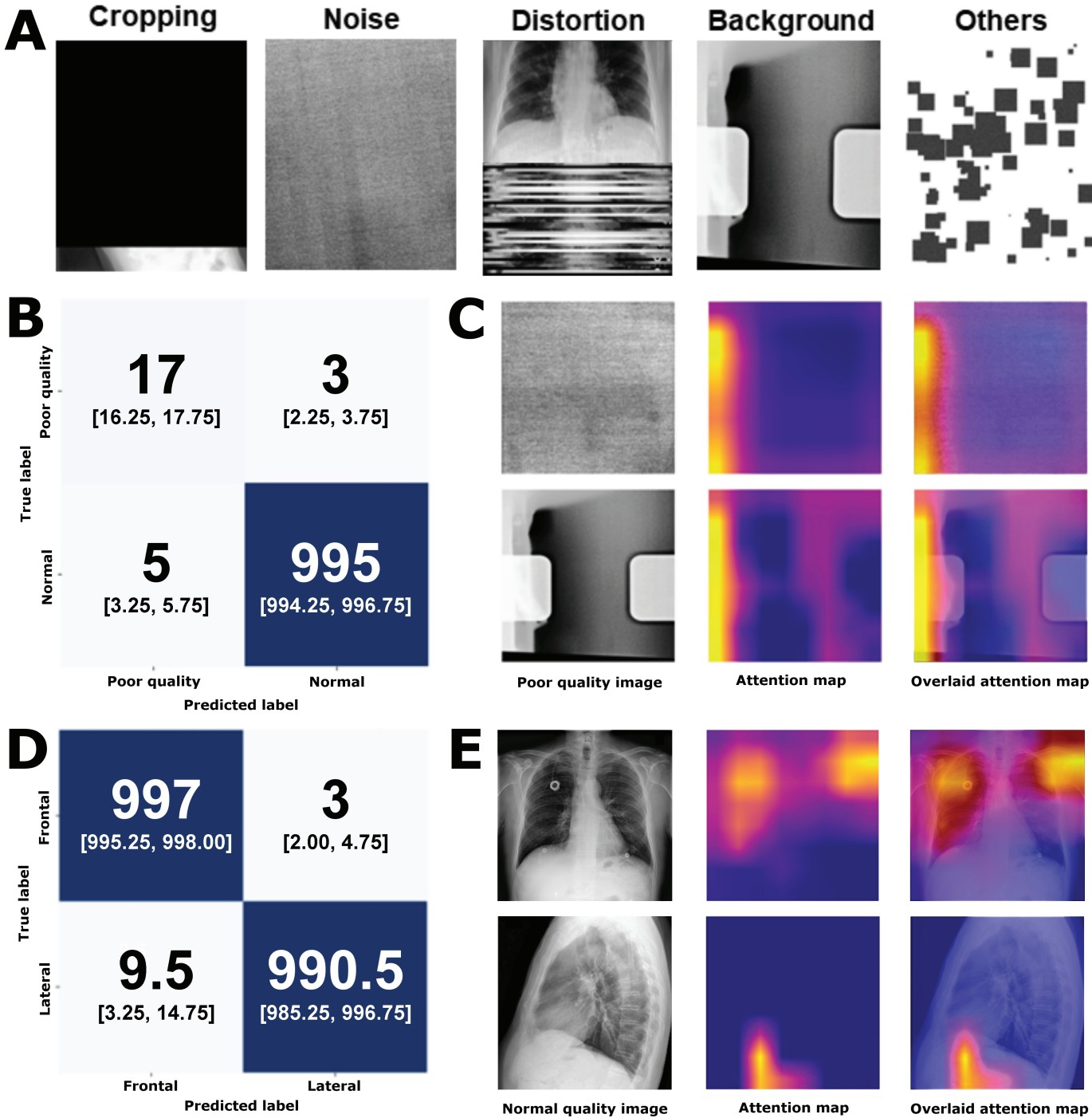

**Fig 3. Data description, computer vision model classification results, and model interpretability.** (A) Generated poor-quality images due to cropping, noise, distortion, background and other types (patching) of image manipulations. (B) Confusion matrix of normal versus poor quality classification model with the median [interquartile range (IQR)] displayed across 10 models initialised with different random states. (C) Example of poor-quality input images and the model attention map. (D) Confusion matrix of frontal versus lateral classification model (median [IQR] across 10 models initialised with different random states). (E) Example of frontal (from [31]) and lateral (from [32]) input images and the model attention map.

For both sub-models (see Fig 2), model explainability was explored through mapping model attention with Grad-CAM (Fig 3C and 3E). Model attention for sub-model 1 is relatively localised with a region of high intensity on the left. For sub-model 2, model attention is diffused when frontal images are encountered, covering most of the image with higher attention zones covering both lungs. When presented with a lateral image, model attention is markedly more focused and often localised to the region just below the diaphragm.

## Cardiomegaly severity classification

In MIMIC-CXR, 'Normal' is the most common descriptor for heart size with 10,883 cases, followed by 'Mild' (4,651 cases), 'Borderline' (4,172 cases) and 'Moderate' (2,392 cases) when describing positive cardiomegaly. Generally, descriptor frequency decreases with increasing perceived severity. 'Mild to moderate', 'Moderate to severe' and 'Severe', are less commonly used, each with less than a thousand cases (Fig 4A).

After CV-CTR calculation, severity descriptors were ranked by median CV-CTR and CAR. For each pair of neighbouring descriptors, Mann-Whitney U-tests were performed to identify statistically synonymous severity descriptors which were then merged to form seven severity categories.

For each descriptor, CV-CTR was calculated for all corresponding PA CXR images with the CV model (Fig 4A). The ranking of descriptors by median is mostly in line with linguistic expectations. 93.8% of mild cardiomegaly cases were found to have CV-CTR over the clinical threshold of 0.5 based on the CV model, with upwards to over 98% in severe cardiomegaly, indicating high recall of the CAD system for cardiomegaly cases.

CV-CTR intervals for each severity group were subsequently established for each severity category (Fig 4B). The upper threshold for 'normal' CTR (0.51) is close to 0.5, mirroring the clinical threshold for cardiomegaly diagnosis. The ratio intervals for normal and severe categories are much larger than intermediates to encompass extreme values. The number of cases falling into each severity interval varies slightly compared to the reference distribution of cases based on original report descriptors.

Using previously established severity group CV-CTR intervals, a confusion matrix was constructed to evaluate heart size predictions of the model against MIMIC 'ground truth' labels (Fig 4C). The model effectively assessing cardiomegaly severity at both ends of the severity spectrum. Intermediate severity categories, such as 'moderate', saw a drop in accuracy in severity classification, although the model is still highly accurate for binary classification (normal vs cardiomegaly) across all categories excluding 'borderline'.

The severity classification based on CV-CTR is similar to the CAR, defined here as the ratio of the total heart area and lung area as presented in PA CXR images. The distribution pattern of CV-CAR closely resembles that of CV-CTR. They are found to be positively correlated ($R^2$=0.82), with 90% of residuals within $\pm$0.09. Additionally, the CV-CAR exhibits consistent patterns across the 7 severity categories (S1 Fig).

Once a severity category prediction has been made by the CV model, it is used as an input argument for report sentence generation using LLMs. A sentence is picked at random from the pre-curated list of generated sentences of the corresponding severity category to constitute the final report sentence. Fig 5 illustrates three examples of report sentence generation based on cardiomegaly severity predictions. Model prediction for the CXR image in Fig 5A matches that of the original report, and as such a sentence resembling the original is generated. Model prediction for the CXR image in Fig 5B differs from the original by one level of severity and thus is regarded as a close match, while a mismatch is found for the CXR image in Fig 5C.

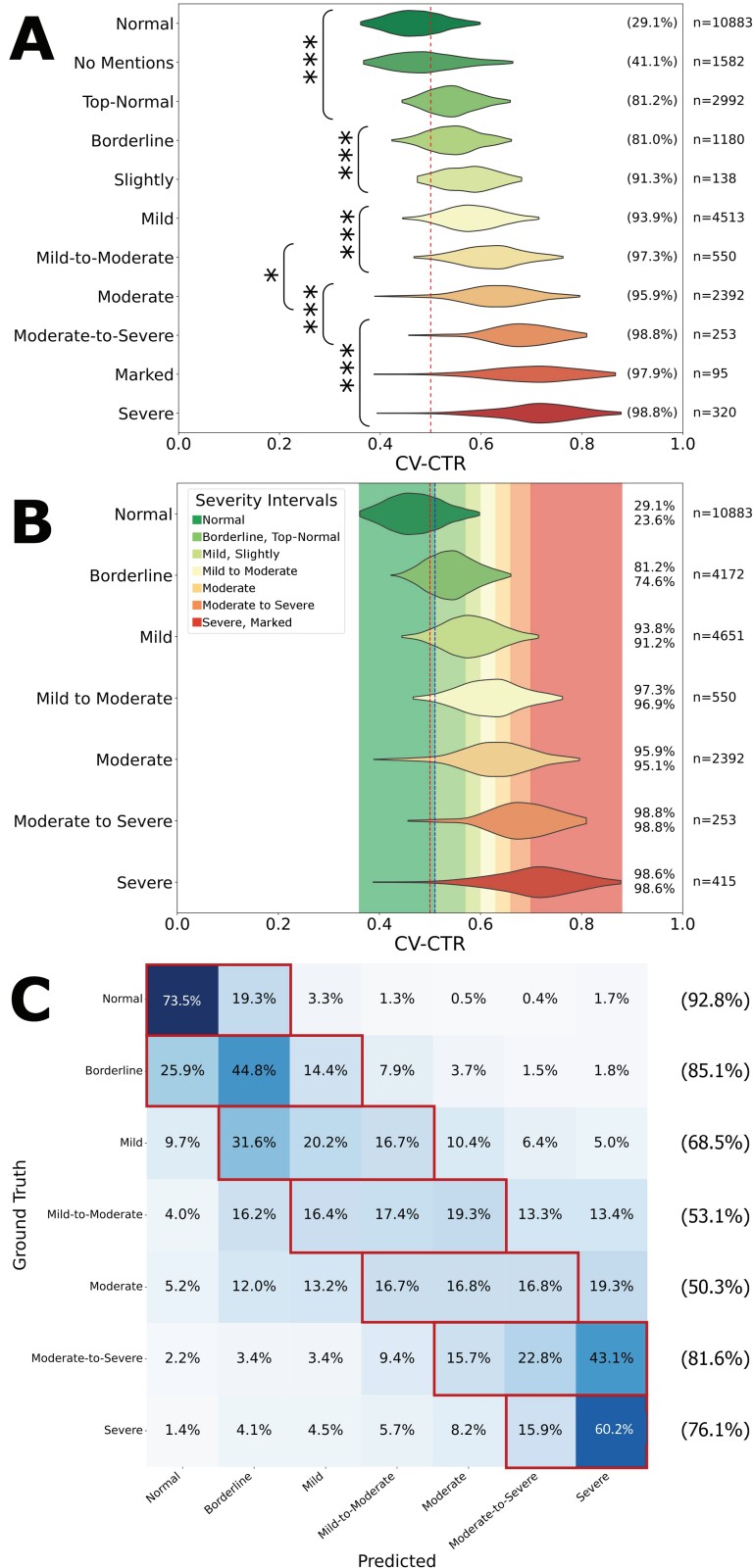

**Fig 4. Data-driven severity classification based on natural language extracted labels and computer vision-based cardiothoracic ratio calculation.** A: Violin plot of the cardiothoracic ratio (CTR) based on the computer vision (CV) segmentation model. Each adjective/adverb relating to cardiomegaly was normalised (i.e. 'mild*ly*' to 'mild').

Also showing reports without cardiomegaly mentions. Comparison of CV-CTR across different severity indicators. * = P<0.05, *** = P<0.001. The percentage on the right-hand side indicates the proportion of images with a CTR over 0.5 (red dashed line). B: Cardiomegaly severity classification model with thresholds determined by the model. Severity indicators with non-significant different CV-CTRs between them were grouped. The blue line indicates the cut-off for which the probability of borderline cardiomegaly is larger than normal heart size. The top percentage on the right-hand side indicates the proportion of images with a CTR over 0.5 (red dashed line), whereas the bottom percentage indicates the proportion of images with a CTR over 0.51, the threshold between normal and borderline heart size (blue dashed line). C: Confusion matrix of the CV-CTR classification model into the severity classes. Red boxes and percentages indicate the total number predicted within one category of the labels from radiology reports.

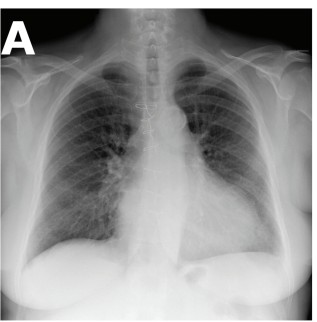

**Report severity:**
Borderline
**Predicted severity:**
Borderline (CV-CTR=0.54)
**Synthetic report sentence:**
"Borderline enlarged heart size."
**Generated sentence:**
"Cardiac size is top-normal."

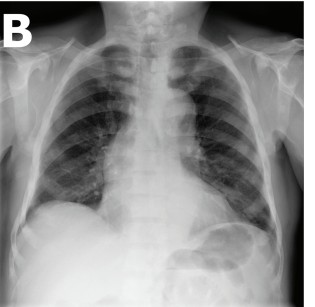

**Report severity:**
Normal
**Predicted severity:**
Borderline (CV-CTR=0.54)
**Synthetic report sentence:**
"Heart size is within normal limits."
**Generated sentence:**
"Top normal heart size."

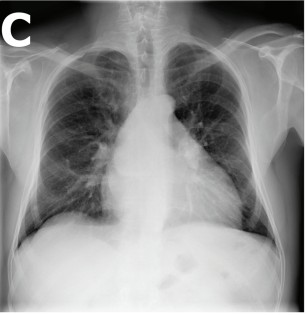

**Report severity:**
Moderate
**Predicted severity:**
Normal (CV-CTR=0.44)
**Synthetic report sentence:**
"Moderate cardiomegaly stable."
**Generated sentence:**
"Heart size appears normal."

**Fig 5. Examples of computer vision-based cardiothoracic ratio calculation and subsequent report text generation.** (**A**) Exact match (image from [31]; see MIMIC report s56175428 for original report sentence and associated image). (**B**) Close match (within 1 category of report text) (image from [32]; original MIMIC report s52317659). (**C**) Mismatch between CAD system and image (image from [31]; original MIMIC report s57481090).

## Human evaluation of radiology reports

We calculated the strict and relaxed agreement in three different settings: (1) between our radiologists (strict: 0.487, relaxed: 0.864), (2) between our radiologists and the original MIMIC radiologists on unchanged sentences (strict: 0.624, relaxed: 0.845), and (3) between our radiologists and machine-generated sentences (strict: 0.545, relaxed: 0.929).

Notably, the machine-generated sentences achieved a strict agreement of 0.545, which was 0.089 lower than the maximum strict agreement between our radiologists and the MIMIC radiologists (0.634). In contrast, the machine-generated sentences achieved the highest relaxed agreement (0.929), surpassing the agreement between our radiologists and the MIMIC radiologists by 0.084.

We evaluated the relationship between other conditions in the reports and cardiomegaly. We show here the most common co-occurrences across all reports, and for those with co-occurrences with cardiomegaly (Fig 6A) we visualise the CV-CTR in Fig 6B. Between 53% (lung lesion) and 90% (oedema) of patients had these as comorbidities with presence of cardiomegaly.

## Discussion

The presence of quality issues, such as poor quality images and mislabelling within the MIMIC-CXR dataset, is an unexpected finding despite the relative rarity. Regardless of the underlying causes, if not addressed and unnoticed by researchers, the quality issues could pose significant challenges to ensure reliability of future studies utilising the dataset. Current literature involving the dataset rarely mentions the dataset quality shortcomings, nor do they address it. The original article describing the MIMIC-CXR dataset does contain a brief notice of the variability of images caused by image rotation, poor patient positioning and secondary collimation [33]. However, it did not comment on the presence of severe image quality issues nor instances of mislabelling. Out of five studies published over the last few years involving the use of MIMIC-CXR images for creating deep learning models [40–44], none has acknowledged any aspects of quality issues within the dataset.

We aimed to identify and address certain quality issues in MIMIC-CXR to assist future studies using the dataset. Manual identification of bad-quality images or wrong view labels is possible for smaller datasets, but the volume of images in MIMIC-CXR makes this process practically impossible. The ResNet50 model 1 from the study was used to identify potentially poor quality images. All images classified as poor quality by the model were reviewed by two humans, and resulted in a total of 83 images with severe quality issues that are indicated as such in our meta data file. We applied the same strategy (human evaluation after model indicates a mislabelled view) using model 2, with a total of 1,054 images whose view was corrected. We then applied this model to classify all images without view labels and manually checked these. We acknowledge that the ResNet50 models can be further improved. Judging from model attention heatmaps, model 1 frequently displays a stronger focus around the edge of the image as opposed to inner zones, which may limit its ability to identify certain types of poor quality images. The hierarchical nature of our framework makes model 1 a potential weak spot, as its non-zero false negative rate means a poor quality image can proceed to view classification. However, since poor quality images are rare, the overall impact is likely limited. The augmented image metadata dataset (S1 Data) represents an improvement over the original, serving to maximise the exclusion of poor quality images and allow images with the correct views to be used in future studies. Here we only used frontal images (PA/AP) for cardiomegaly detection, and did not use any image classified as lateral.

Our choice for cardiomegaly as a use case was motivated by the fact that in reports cardiomegaly often appears by itself in a sentence. No other observations are commonly reported in the same sentence with cardiomegaly. This is unlike other potential use cases such as pneumonia (occurring with mentions of opacities), or focal consolidation, pleural effusion and pneumothorax that are often mentioned in the same sentence when findings for these 3 are negative. Our CV-based model was used to predict the severity of cardiomegaly, and thereby

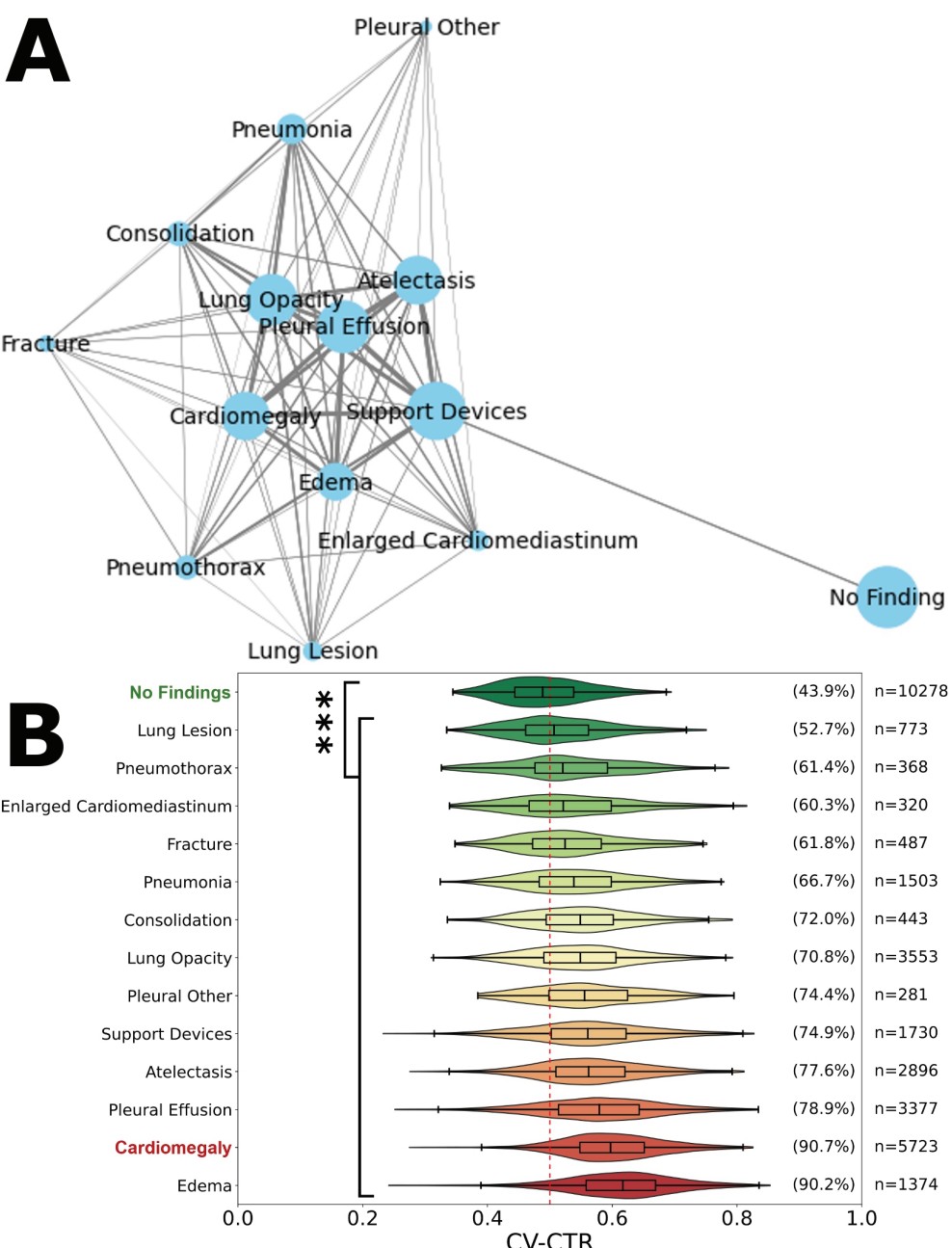

**Fig 6. Evaluation of the cardiothoracic ratio and comorbidities with cardiomegaly.** (**A**) Co-occurrence graph of the other 14 CheXpert observations by report, including cardiomegaly. Node size indicates the number of reports where the condition is observed. Line width indicates the association between conditions measured by Ochiai similarity. (**B**) Violin plot of computer vision-based cardiothoracic ratio for the most common co-occurrences with other mentions. The percentage on the right-hand side indicates the proportion of images with a CTR over 0.5 (red dashed line). Significant differences from the reference group noted, ***P<0.001.

also binary classification. Cardiomegaly is often a manifestation of dilated or hypertrophic cardiomyopathies [23,45,46]. While the aetiology of both cardiomyopathies is highly complex and not yet fully elucidated, it is known that both could lead directly to heart failure syndrome

which carries a poor prognosis [23,47,48]. A CAD system that performs as well as human readers may have clinical utility in future.

The segmentation model was not trained on the MIMIC-CXR dataset for segmentation, therefore all images within the MIMIC-CXR dataset could serve as part of the test set. Our approach has an overall accuracy of 95.2% (0.95 AUC) when evaluated on all images with unequivocal heart size mentions in MIMIC-CXR reports. We used the MIMIC-CXR report diagnosis as ground truth and 0.51 CV-CTR as the threshold for cardiomegaly (borderline cases are excluded as they do not fit into either category). Our CAD model has a false negative rate of 6.6%, indicating that it captures the majority of cardiomegaly cases, while its false positive rate is 3.2% is reflective of high precision of cardiomegaly detection. Also, not all cardiomegaly diagnoses in MIMIC-CXR are necessarily correct, as evidenced below. In terms of overall accuracy, our model proves to be equivalent or superior to nearly all existing classification-based [25] and segmentation-based [24,27,29] models (Table 3) trained on a variety of datasets.

While previous studies have developed similar CAD systems for binary cardiomegaly detection, they have not developed functions that distinguish varying severity levels. In contrast, when radiologists report findings of cardiomegaly, it is often standard practice to attach a severity descriptor. Our work stands out by identifying the most used descriptors, grouping synonyms through statistical analysis, and establishing CV-CTR intervals for each severity category. These advancements make the CAD output more informative and crucial for generating report sentences.

Our work also explores the subjectivity in severity attribution in clinical practice. Without any universal guidelines for descriptor selection, radiologists rely on individual judgment to pick the most appropriate descriptors. This is a process that permits a significant degree of subjectivity, which is reflected by the wide distributions of CV-CTR across different descriptors and severity categories (Fig 4B). While fluctuations in segmentation and CV model performance contribute to the overlaps, the subjective nature and personal biases of radiologists are also likely factors. Furthermore, responses gathered from the radiologist questionnaire revealed 4 cases where the responding radiologist rejected the original MIMIC-CXR judgement of cardiomegaly, which constitutes 12.5% of all original cardiomegaly sentences assessed. In nearly half (47.5%) of all cardiomegaly sentences included in the questionnaire, the two assessing radiologists disagreed in their response agreement levels. Indeed, the strict agreement between assessing radiologists is only 0.488 (0.864 relaxed), indicating a considerable degree of subjectivity in CXR interpretation and reporting. This related mostly to the category of severity, but in one case to the sentence itself which one radiologist considered

**Table 3. Comparison of binary cardiomegaly classification performance between existing models (as reported in publication) and model in the current study. AUC = Area Under the (ROC) Curve. * indicates that the value represents the performance of the best-performing model within the study or the performance measured on a specific test set where the model is most accurate. The top result is indicated in bold, second best underlined.**

| Study | Test data | Size | AUC | Accuracy (%) |
|-------|-----------|------|-----|--------------|
| [25] | ChestX-ray8 | 600 | 0.87 | 79 |
| [29] | Local hospital | 418 | **0.96*** | 91 |
| [27] | Local hospital | 183 | N/A | 94.9 |
| [24] | Local hospital | 131 | 0.92* | 92.3 |
| Ours | MIMIC-CXR | 19,144 | 0.95 | **95.2** |

"too verbose". Nonetheless, a point of positivity lies in the fact that the upper threshold of normal CV-CTR determined by statistical analysis (0.51) is nearly identical to the currently used clinical threshold (0.5), indicating a high level of adherence in clinical practice.

CTR has limited diagnostic power when compared to cardiac MRI, the gold standard for cardiac chamber enlargement detection, according to some studies [49,50]. Therefore, we investigated the CAR as an alternative metric for cardiomegaly detection. Although not used clinically, this concept considers the definition and presentation of cardiomegaly by advantageously considering two heart dimensions instead of the single dimension used in CTR calculations. Despite these advantages, we did not observe a major difference between CTR and CAR in terms of alignment with cardiomegaly severity. However, the CAD system's reliability could be enhanced by cross-checking CV-CTR and CV-CAR from the same images, identifying mismatches in predicted severity between both methods. We also examined correlations between cardiac enlargement and various comorbidities. The mention of oedema appeared to be the strongest predictor for enlarged cardiac size, with CV-CTR greater than 0.5 in over 90% of cases, only marginally less than mentioning of cardiomegaly itself. As such, the presence oedema may be strongly predictive of the co-occurrence of cardiomegaly. This reflected the outcome of the Ochiai co-occurrence analysis showing a frequent co-occurrence between the two conditions (Fig 6B).

The inclusion of an automatic report sentence generation module is another unique feature in a cardiomegaly CAD system. Radiologists' workloads have increased substantially and they have limited time (<10 min) to write reports[51]. Radiologists spend 5-10 seconds on each image[9], highlighting they primarily spend their time writing reports. Therefore, we explored whether we can generate report-style sentences based on the CV model output, and had four radiologists assess the text quality. Compared to more diverse and complex conditions, the reporting of cardiomegaly is generally consistent and uncomplicated, enabling LLMs to imitate their style effectively. This also means that locally deployed LLMs could be used within the secure environment of a hospital trust (i.e. no data leaves the site). By using few-shot learning or fine-tuning approaches, LLMs can generate sentences that imitate individual radiologists' styles. Such sentences can then be suggested to the radiologist when writing the report opposed to a rule-based system which may contain sentences that this radiologist does not typically use. While this is not the scope of this work, such integration could be implemented within local software to accelerate report writing by suggesting sentences that a radiologist can either include in their report or edit as needed. The radiologist's decision to use a specific sentence and whether they choose to edit it afterward can also serve as part of a reinforcement learning by human feedback approach to enhance such a system.

Another strength of the study is the inclusion of radiologist assessment of the performance of the CAD system, a rare inclusion with parallels only found in one other study [27]. Whereas most other work in the space of report generation use natural language generation metrics to evaluate the agreement between generated text and original reports [52–54]. While the assessment is limited in scale, it nevertheless helped validate the system's feasibility. The current CAD system reached an average agreement of S:0.545/R:0.929 with our radiologists' verdicts, comparable with S:0.624/R:0.845 between our radiologists and MIMIC. This suggests that the CAD system's performance may be non-inferior, or share a comparable level of subjectivity, to trained radiologists. Our findings indicate that our machine-generated sentences were on par with the MIMIC radiologists in assessing the severity of cardiomegaly and the semantics used to generate the report. Disagreements are primarily linguistic rather than diagnostic, as evidenced by the reasons for disagreements. These can be addressed by improving the quality of generated sentences.

## Limitations and future work

While the preliminary results are encouraging, the study remains a proof-of-principle at this stage due to the absence of large-scale clinical validations. The system is yet to be tested on datasets other than MIMIC-CXR, and thus direct comparisons of performance with existing models is challenging. While we compare our model to prior work, they did not release their data or models, thereby preventing direct comparison or limiting the assessment of our models' generalisability. To the best of our knowledge, no other large datasets with English-language reports are publicly available. Therefore, there may be gap between the accuracy we report and 'real-world' performance (including different co-morbidities and clinical settings) of such an algorithm. Nevertheless, the test set used in this study (n=19,144) also vastly outnumbers those used in similar studies (131-600) by orders of magnitude, covering a range of disease phenotypes and comorbidities. Furthermore, the per-case performance of the CV model relies on accurate anatomical segmentations, making it prone to errors due to aberrant segmentations. This is particularly concerning in CXR images where either the cardiac silhouette or part of the lungs are obscured by conditions like pulmonary oedema and pleural effusion. These conditions can reduce the reliability of segmentations from algorithms trained with limited data on comorbidities. A more versatile segmentation model optimised for challenging CXR images could greatly benefit the performance of the model. In essence, while the CAD system has considerable potential for clinical deployment with further development, it is not the intention of this study to use it as such in its current state. Therefore, regulatory frameworks and legal aspects will not be discussed.

As PA is the recommended view for cardiomegaly detection, AP CXR images have been excluded from the cardiomegaly analysis. Despite the close resemblance of AP to PA CXR images, AP images tend to exaggerate the heart size [55]. However, AP images can serve as an initial screening for determining the need for further investigation in future. For instance, one study has recommended a CTR ratio of 0.6 as the threshold when using AP CXR for increased specificity [56].

During the preparatory phase in LLM sentence generation, we noted that a significant portion of cardiomegaly mentions include references to past time points (i.e. 'mild cardiomegaly is unchanged', 'severe cardiomegaly has improved'). We, like others, did not explore the possibility of generating sentences capturing multiple time points. However, it may be feasible by allowing the model to calculate CTR and make a severity prediction for both the past and present CXR in question, using any difference as an indicator of change to generate more dynamic sentences. We envision extending our approach to cover additional medical conditions requiring chest X-rays. Such a vision system would output results as a vector and, combined with retrieval-augmented generation (RAG) or in-context learning scenarios, can enable report generation based on similar cases. This would enhance the system's ability to produce relevant, context-aware (including prior time points) clinical reports.

## Conclusion

The present study has developed the essential framework and components for the assembly of a highly automated and reliable CAD system for cardiomegaly detection using PA CXR images. The system holds several advantages, including a high degree of cost-effectiveness, explainability in detection, and efficiency in operation. Input CXR images are first screened to exclude and flag instances unsuitable for cardiomegaly detection. By calculating CTR using a CV model from deep learning-derived heart and lung segmentations, high overall accuracy in detecting cardiomegaly and assigning appropriate severity labels is achieved. Furthermore,

the model is capable of generating human-like, report-styled sentences based on the output of the CV model. Performance metrics suggest that the model's ability to produce accurate cardiomegaly report sentences using PA CXR is likely non-inferior to human radiologists. Thus, further studies are needed for confirmation. While the current model remains a proof-of-concept and is not deployed in clinic, it can serve as a framework for others to further optimise or fine-tune these models and evaluate their performance on local data. Upon validation in this way, the novel CAD system could represent an accessible and cost-effective framework providing radiologists with a second opinion for validating their assessments.

## Acknowledgments

We gratefully acknowledge the support of Dr Catherine Payne, Dr Abi Redwood, Dr Liam Roebuck and Dr John Taylor (listed alphabetically) in scoring the reports. We would also like to express our gratitude to Dr Jamie Campbell and Dr Joanna Davis for their time and assistance in gathering expert feedback. We thank Dr Tom Pollard and Dr Alistair Johnson for their advice and clarifications on the MIMIC-CXR data user agreement that prohibits disclosure of data (images, report snippets). The NIH Clinical Center was the provider of the de-identified images of chest x-rays shown in this manuscript, and is available through the NIH download site: https://nihcc.app.box.com/v/ChestXray-NIHCC.

## Supporting information

**S1 Fig. Data-driven severity classification based on cardiothoracic area ratio.** Data-driven severity classification based on natural language extracted labels and computer vision-based cardiothoracic area ratio (CAR) calculation. (**A**) CV-CAR vs CV-CTR with the 7 severity groups from reports indicated by colours. The horizontal red dashed line represents CV-CTR = 0.50 (common threshold for cardiomegaly), the vertical red dashed line represents the CAR value for the lowest 1% of all cardiomegaly cases (from mild to severe), with 99% of cardiomegaly cases to be found above this threshold. (**B**) Residual plot based on the line of best fit. The parallel black dashed lines represents the interval where 90% of residuals reside. (**C**) Violin plot of the CV-CAR for all categories prior to merging categories. (**D**) Violin plot of the CV-CAR after consolidation of categories. (C and D) The red dashed line represents the 1% CAR value threshold for cardiomegaly cases [from mild to severe, as in (A)].
(TIF)

**S1 Data Updated meta data file for MIMIC-CXR dataset (377,110 images).** This meta data file contains original data of the dicom_id, subject_id, study_id, and original view position, and adds data on image quality (TRUE/FALSE), corrected view (FRONTAL/LATERAL), identified sex differences (TRUE/FALSE), view agreement of original and new labels (TRUE/FALSE), the test set IDs for the view correction, quality assessment and cardiomegaly classification, and borderline cardiomegaly instances (TRUE/FALSE).
(CSV)

## Author contributions

**Conceptualization:** Matt Grech-Sollars, Joram M. Posma.

**Data curation:** Tianhao Zhu.

**Formal analysis:** Tianhao Zhu, Kexin Xu, Wonchan Son.

**Methodology:** Tianhao Zhu, Kexin Xu, Antoine D. Lain, Joram M. Posma.

**Supervision:** Kristofer Linton-Reid, Marc Boubnovski-Martell, Matt Grech-Sollars, Antoine D. Lain, Joram M. Posma.

**Visualization:** Tianhao Zhu, Kexin Xu, Joram M. Posma.

**Writing – original draft:** Tianhao Zhu, Kexin Xu, Matt Grech-Sollars, Antoine D. Lain, Joram M. Posma.

**Writing – review & editing:** Kristofer Linton-Reid, Antoine D. Lain, Joram M. Posma.

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
