## [Decision Letter · Decision Letter 0]

13 Nov 2024

PDIG-D-24-00379Designing a computer-assisted diagnosis system for cardiomegaly detection and radiology report generationPLOS Digital HealthDear Dr. Posma, Thank you for submitting your manuscript to PLOS Digital Health. After careful consideration, we feel that it has merit but does not fully meet PLOS Digital Health's publication criteria as it currently stands. Therefore, we invite you to submit a revised version of the manuscript that addresses the points raised during the review process. Please submit your revised manuscript within 60 days Jan 12 2025 11:59PM. If you will need more time than this to complete your revisions, please reply to this message or contact the journal office at digitalhealth@plos.org. Please include the following items when submitting your revised manuscript:* A rebuttal letter that responds to each point raised by the editor and reviewer(s). You should upload this letter as a separate file labeled 'Response to Reviewers'. This file does not need to include responses to any formatting updates and technical items listed in the 'Journal Requirements' section below.* A marked-up copy of your manuscript that highlights changes made to the original version. You should upload this as a separate file labeled 'Revised Manuscript with Track Changes'.* An unmarked version of your revised paper without tracked changes. You should upload this as a separate file labeled 'Manuscript'. If you would like to make changes to your financial disclosure, competing interests statement, or data availability statement, please make these updates within the submission form at the time of resubmission. Guidelines for resubmitting your figure files are available below the reviewer comments at the end of this letter. We look forward to receiving your revised manuscript.

Kind regards.

Hualou Liang

Academic EditorPLOS Digital Health

Hualou Liang

Academic EditorPLOS Digital Health Leo Anthony CeliEditor-in-ChiefPLOS Digital Healthorcid.org/0000-0001-6712-6626**Journal Requirements:**

1. We ask that a manuscript source file is provided at Revision. Please upload your manuscript file as a .doc, .docx, .rtf or .tex.

2. We have amended your Competing Interest statement to comply with journal style. We kindly ask that you double check the statement and let us know if anything is incorrect. 

 **Additional Editor Comments (if provided):****Reviewers' Comments:**Reviewer's Responses to Questions

**Comments to the Author**

1. Does this manuscript meet PLOS Digital Health’s publication criteria? Is the manuscript technically sound, and do the data support the conclusions? The manuscript must describe methodologically and ethically rigorous research with conclusions that are appropriately drawn based on the data presented.

Reviewer #1: Partly

Reviewer #2: Yes

Reviewer #3: No

2. Has the statistical analysis been performed appropriately and rigorously?

Reviewer #1: I don't know

Reviewer #2: Yes

Reviewer #3: No

3. Have the authors made all data underlying the findings in their manuscript fully available (please refer to the Data Availability Statement at the start of the manuscript PDF file)?

Reviewer #1: Yes

Reviewer #2: Yes

Reviewer #3: Yes

4. Is the manuscript presented in an intelligible fashion and written in standard English?

Reviewer #1: Yes

Reviewer #2: Yes

Reviewer #3: No

5. Review Comments to the Author

Reviewer #1: Overall, the manuscript presents a promising approach to utilizing computer-assisted diagnosis (CAD) for cardiomegaly detection and radiology report generation, achieving impressive accuracy. The research is technically sound and addresses an important problem in radiology, particularly in reducing radiologist workloads.

1. Technical Soundness and Data Support:

The study demonstrates a well-structured methodology with clear data to support its findings. The use of the MIMIC-CXR dataset is appropriate, and the system’s performance, particularly in terms of accuracy (95.2%) and report generation consistency, is commendable. However, some additional details on how the dataset inconsistencies were corrected would enhance transparency.

2. Statistical Analysis:

While the manuscript focuses primarily on the model’s accuracy, there is limited mention of detailed statistical analysis. It would be helpful to include more information on the statistical methods used to validate the model’s performance, such as confidence intervals or statistical significance testing, to ensure robustness.

3. Data Availability:

The use of the publicly available MIMIC-CXR dataset complies with PLOS’s data availability policy. However, the authors should ensure that all data underlying their findings, including any processed or additional datasets, are available in a public repository or provided as supplementary material.

4. Language and Presentation:

The manuscript is generally well-written and presented in standard English, with only minor grammatical or typographical errors that can be addressed during revisions. The content is clear and understandable, making it accessible to a broad audience.

5. Improvements and Future Directions:

The authors mention future improvements to the text generation aspect of the CAD system. It would be beneficial to include more specific details on how these enhancements will be approached. Additionally, a more in-depth explanation of how the system corrects inconsistencies in the dataset would strengthen the manuscript.

6. Recommendation:

Overall, I recommend minor revisions to address the points mentioned above, particularly the need for more detail on the statistical analysis and future improvements.

Reviewer #2: The study presents a compelling, well-structured CAD system that addresses a significant need in radiology by automating cardiomegaly detection and report generation. The use of a large, publicly available dataset (MIMIC-CXR) and the system's high accuracy rates make it a strong contribution to the field. The authors have effectively combined computer vision with NLP to create a practical solution for radiologists, supported by validation from certified radiologists.

Additional clarity on how explainability was ensured in decision-making, particularly in areas where the model diverged from human interpretations, would be beneficial.

Reviewer #3: The present paper proposes building a system in which x-ray images are processed and put through a pipeline to classify cardiomegaly and then produce a written radiology report. The novelty and effectiveness of the proposed system are unclear from the paper. The lack of clarity is in part due to the presentation and writing of the paper. The paper may be suitable for publication after the following major issues are addressed.

Major comments:

The novelty of the paper is lacking, and the objective is not clear. The abstract and introduction need to be rewritten to lay out the structure and define the novelty. The methodology and results need to be rewritten to present the information linearly and logically.

The paper fails to compare with prior work.

The clinical application is unclear. For example, does the method work for a screening application with an asymptomatic population? Why would people be screened for cardiomegaly and not also for pneumonia and other diseases? If only for cardiomegaly, then how are incidental findings handled?

When Resnet is trained, it is not clear what “filtered images” are.

The segmentation algorithm should be compared to outlines of the heart created by radiologists.

The performance of the model is evaluated solely on the MIMIC-CXR dataset. There is no external validation using independent datasets to assess the generalizability of the model. Without testing on other datasets, it is unclear whether the model's performance holds in different clinical settings or with data from other sources.

The manuscript claims a state-of-the-art accuracy of 95.2% for binary cardiomegaly classification. However, this claim is based on comparisons with studies that may have used different datasets, evaluation metrics, or methodologies. Without benchmarking the model against existing methods using standardized datasets and protocols, the claim is not substantiated.

Why is Natural Language Processing used to generate the report instead of a rule-based system?

The paper spends extensive time discussing the errors in the dataset and how they were identified (Table 2). However, the denominator of the total inaccuracies is not identified, indicating that there are even more errors in the dataset.

ROC, precision, and recall should be reported. Estimates of error should be calculated and reported with all performance metrics.

The paper's statistical methods have several concerns. The use of multiple t-tests with Šidák correction to identify statistically synonymous severity descriptors is problematic. Data skewness and unequal variances violate key t-test assumptions. Assigning threshold values by selecting the highest p-value from t-tests at incremental intervals is not a standard method for cut-off determination in continuous data. This approach increases the risk of Type I errors and does not adequately handle issues from multiple comparisons.

The methodology should be rewritten to enable reproducibility. Currently, methodology and implementation details are missing or unclear. Unnecessary implementation details should be moved to appendix or removed.

Minor comments:

The use of adjectives and adverbs introduces ambiguity into the paper. The use of compounded sentence structure hinders the readability and clarity of the paper. Adjectives and adverbs should be deleted or replaced with quantifiers. Compound and complex sentence structures with mixed clauses should be simplified and presented linearly. The issue is most evident in the abstract and introduction.

Heading and subheadings are overused and misused throughout the paper. The current use of subheadings hinders readability and obscures the continuity and logic between the paragraphs. The number of subheadings should be reduced and the paragraphs should be connected cohesively to each other.

Figures are misplaced in the middle of paragraphs and not discussed in the paper.

The input of the dataset into MedQBot may violate the IRB of the dataset.

The paper briefly mentions future improvements but fails to discuss current limitations in depth, such as the model's inability to handle images with other pathologies that affect heart size or lung fields.

“Our computer vision model, developed as part of the CAD system, exhibited good 437 performance for binary cardiomegaly classification.” “Good” is an unsubstantiated adjective.

The discussion should be expanded to include a discussion on the gap between validation metrics and real-world performance.

The paper lacks a discussion on the legal implications of deploying an autonomous CAD system and replacing/augmenting a radiologist in clinical settings, including regulatory approvals and compliance with healthcare laws.

6. PLOS authors have the option to publish the peer review history of their article (what does this mean?). If published, this will include your full peer review and any attached files.

**Do you want your identity to be public for this peer review?** For information about this choice, including consent withdrawal, please see our Privacy Policy.

Reviewer #1: No

Reviewer #2: No

Reviewer #3: No

---

## [Decision Letter · Decision Letter 1]

3 Mar 2025

PDIG-D-24-00379R1Designing a computer-assisted diagnosis system for cardiomegaly detection and radiology report generationPLOS Digital Health Dear Dr. Posma, Thank you for submitting your manuscript to PLOS Digital Health. After careful consideration, we feel that it has merit but does not fully meet PLOS Digital Health's publication criteria as it currently stands. Therefore, we invite you to submit a revised version of the manuscript that addresses the points raised during the review process. Please submit your revised manuscript within 30 days Apr 02 2025 11:59PM. If you will need more time than this to complete your revisions, please reply to this message or contact the journal office at digitalhealth@plos.org. Please include the following items when submitting your revised manuscript:* A rebuttal letter that responds to each point raised by the editor and reviewer(s). You should upload this letter as a separate file labeled 'Response to Reviewers'. This file does not need to include responses to any formatting updates and technical items listed in the 'Journal Requirements' section below.* A marked-up copy of your manuscript that highlights changes made to the original version. You should upload this as a separate file labeled 'Revised Manuscript with Track Changes'.* An unmarked version of your revised paper without tracked changes. You should upload this as a separate file labeled 'Manuscript'. If you would like to make changes to your financial disclosure, competing interests statement, or data availability statement, please make these updates within the submission form at the time of resubmission. Guidelines for resubmitting your figure files are available below the reviewer comments at the end of this letter. We look forward to receiving your revised manuscript. Kind regards, Hualou LiangAcademic EditorPLOS Digital Health Hualou LiangAcademic EditorPLOS Digital Health Leo Anthony CeliEditor-in-ChiefPLOS Digital Healthorcid.org/0000-0001-6712-6626 **Additional Editor Comments (if provided):****Reviewers' Comments:** Reviewer's Responses to Questions

**Comments to the Author**

1. If the authors have adequately addressed your comments raised in a previous round of review and you feel that this manuscript is now acceptable for publication, you may indicate that here to bypass the “Comments to the Author” section, enter your conflict of interest statement in the “Confidential to Editor” section, and submit your "Accept" recommendation.

Reviewer #1: All comments have been addressed

Reviewer #3: All comments have been addressed

2. Does this manuscript meet PLOS Digital Health’s publication criteria? Is the manuscript technically sound, and do the data support the conclusions? The manuscript must describe methodologically and ethically rigorous research with conclusions that are appropriately drawn based on the data presented.

Reviewer #1: Yes

Reviewer #3: Yes

3. Has the statistical analysis been performed appropriately and rigorously?

Reviewer #1: Yes

Reviewer #3: Yes

4. Have the authors made all data underlying the findings in their manuscript fully available (please refer to the Data Availability Statement at the start of the manuscript PDF file)?

Reviewer #1: Yes

Reviewer #3: Yes

5. Is the manuscript presented in an intelligible fashion and written in standard English?

Reviewer #1: Yes

Reviewer #3: No

6. Review Comments to the Author

Reviewer #1: The manuscript is well-structured, methodologically sound, and meets the publication criteria of PLOS Digital Health. The research is rigorous, the statistical analysis is appropriately conducted, and the conclusions are well-supported by the data. The manuscript is clearly written in standard English, with no major language issues. The authors have adequately addressed all previous comments, and the data availability requirements have been met. Overall, the submission is solid and suitable for publication.

Reviewer #3: From a technical standpoint, the authors have addressed all my comments.

However, the overall readability of the paper is still poor. The readability has deteriorated because the authors have added material instead of removing and reorganizing content. While the stated objectives are clear, other elements clutter the manuscript and distract from its focus.

The manuscript contains grammatical errors and poor sentence structures that require copyediting. There are adjectives and adverbs that should be removed, along with several subject–verb agreement issues. Specific examples include, but are not limited to, Lines 15, 18, 22, and 486.

The authors should have provided a revised manuscript with track changes. The authors should have included the specific text that was added or revised in the manuscript in their responses to reviewers.

I leave the final decision to the editor. From a purely technical standpoint, the paper is acceptable for publication after copyediting. However, if clarity and readability are among the acceptance criteria, I recommend rejection of the paper in its current form.

7. PLOS authors have the option to publish the peer review history of their article (what does this mean?). If published, this will include your full peer review and any attached files.

**Do you want your identity to be public for this peer review?** For information about this choice, including consent withdrawal, please see our Privacy Policy.

Reviewer #1: No

Reviewer #3: No

---

## [Editor Report · Decision Letter 2]

26 Mar 2025

Designing a computer-assisted diagnosis system for cardiomegaly detection and radiology report generation

PDIG-D-24-00379R2

Dear Dr Posma,

We are pleased to inform you that your manuscript 'Designing a computer-assisted diagnosis system for cardiomegaly detection and radiology report generation' has been provisionally accepted for publication in PLOS Digital Health.

Best regards,

Hualou Liang

Academic Editor

PLOS Digital Health